# Obstructive Sleep Apnea and Acute Lower Respiratory Tract Infections: A Narrative Literature Review

**DOI:** 10.3390/antibiotics13060532

**Published:** 2024-06-06

**Authors:** Marko Nemet, Marija Vukoja

**Affiliations:** 1Faculty of Medicine, University of Novi Sad, 21000 Novi Sad, Serbia; marija.vukoja@mf.uns.ac.rs; 2The Institute for Pulmonary Diseases of Vojvodina, Sremska Kamenica, 21204 Novi Sad, Serbia

**Keywords:** obstructive sleep apnea, respiratory infections, pneumonia, influenza, COVID-19, antibiotics, treatment

## Abstract

Both obstructive sleep apnea (OSA) and acute lower respiratory tract infections (LRTIs) are important global health issues. The pathophysiological links between OSA and LRTIs include altered immune responses due to chronic intermittent hypoxia and sleep fragmentation, increased aspiration risk, and a high burden of comorbidities. In this narrative review, we evaluated the current evidence on the association between OSA and the incidence and outcomes of acute LRTIs in adults, specifically community-acquired pneumonia and viral pneumonia caused by influenza and severe acute respiratory syndrome coronavirus 2 (SARS-CoV-2). Studies have demonstrated that OSA patients are more likely to develop bacterial pneumonia and exhibit a higher risk of invasive pneumococcal disease. The risk intensifies with the severity of OSA, influencing hospitalization rates and the need for intensive care. OSA is also associated with an increased risk of contracting influenza and suffering more severe disease, potentially necessitating hospitalization. Similarly, OSA contributes to increased COVID-19 disease severity, reflected by higher rates of hospitalization, longer hospital stays, and a higher incidence of acute respiratory failure. The effect of OSA on mortality rates from these infections is, however, somewhat ambiguous. Finally, we explored antibiotic therapy for OSA patients with LRTIs, addressing care settings, empirical regimens, risks, and pharmacokinetic considerations. Given the substantial burden of OSA and its significant interplay with acute LRTIs, enhanced screening, targeted vaccinations, and optimized management strategies for OSA patients should be prioritized.

## 1. Introduction

Obstructive sleep apnea (OSA) is the most common sleep-related breathing disorder, affecting nearly one billion people worldwide [1]. The prevalence of OSA has increased over the past decades and exceeds 50% in some countries; still, the majority of OSA patients remain unrecognized and undertreated [1,2]. OSA is characterized by repeated periods of complete or partial collapse of the upper airway that lead to oxygen desaturation or arousal from sleep [3]. This induces chronic intermittent hypoxia (IH), sleep fragmentation, and increased sympathetic nervous system activity, all of which have been shown to increase the risk of chronic organ damage [4,5]. Numerous reports confirm that OSA is associated with an increased risk and worse outcomes of chronic cardiovascular, pulmonary, metabolic, neurological, and mood disorders [6,7].

Acute lower respiratory tract infections (LRTIs) are infections within the respiratory tract below the larynx. In the adult population, LRTIs commonly refer to bacterial and viral pneumonia and bronchitis [8]. LRTIs continue to be a major health issue, causing substantial mortality, morbidity, and economic burden. In 2019, LRTIs claimed 2.6 million lives and were ranked as the fourth leading cause of death globally [9,10]. The recent Coronavirus Infectious Disease 2019 (COVID-19) pandemic has dramatically increased morbidity and mortality due to LRTIs and has raised awareness of viral LRTIs, especially in pandemic settings [11].

The susceptibility to respiratory infections and outcomes of patients with LRTIs are determined by the virulence of the organism and the host immune response [12]. Although many studies have proven that OSA induces a pro-inflammatory state [13], the relationship between OSA and respiratory infections has only recently been called to attention. The coexistence of OSA and chronic respiratory infections such as bronchiectasis and cystic fibrosis is common and may worsen patient outcomes by enhancement of pro-inflammatory stimuli [14]. This relationship goes beyond the association between OSA and chronic respiratory infections, as the increasing body of evidence suggests that OSA may also have an important role in the susceptibility to, pathogenesis of, and outcomes of acute respiratory infections. Given the high burden of both OSA and acute LRTIs in the general population, their potential link is critical to explore.

This review aims to evaluate the current evidence on the association of obstructive sleep apnea with the incidence and outcomes of acute lower respiratory tract infections in adults. Specifically, we focus on the relationship between OSA and community-acquired pneumonia (CAP), as well as viral pneumonia caused by influenza viruses and severe acute respiratory syndrome coronavirus 2 (SARS-CoV-2).

## 2. Literature Search Strategy

The search strategy was structured to evaluate the relationship between OSA and bacterial, influenza, and COVID-19 pneumonia. The search was conducted using the PubMed database on 10 January 2024, without any restrictions on the date of publication. The literature search was repeated on 15 April 2024, to screen for any newly published papers that might have been missed by the initial search. The search was conducted using a set of keywords detailed in Table 1. Original research articles that focused on OSA as a potential risk factor for acquiring acute LRTIs or experiencing higher severity or adverse outcomes of these infections were included. Authors excluded publications not available in English, narrative reviews, letters to the editor, and abstracts for which the full text was not available. An overview of the study selection is presented in Appendix A, an online data supplement.

The initial screening process involved evaluating the titles and abstracts of the retrieved articles, which was independently conducted by both authors (MN, MV). In instances where discrepancies arose regarding the inclusion or exclusion of specific studies, the authors had an in-depth discussion to resolve conflicts. During these discussions, authors performed a full-text review of the articles in question to make an optimal informed decision. Additional studies that were not captured through the primary database search were included after manually examining the reference lists of the selected articles. The data extraction process for the selected studies included the authors and date of publication, the title of the study, its aim, design, study groups, inclusion and exclusion criteria, outcomes, an overview of the key findings, and any identified study limitations. This extraction process was divided between the two authors (MN, MV). To guarantee consistency and reliability in the data extraction process, both authors conducted oversight of each other’s data extraction. The summary of the most relevant literature on the association of OSA and LRTIs is presented in Table 2.

## 3. Obstructive Sleep Apnea and Community-Acquired Pneumonia

Our review identified six studies evaluating the risk of contracting and severity of community-acquired pneumonia in OSA patients (Table 2). Out of those six, five studies supported that OSA increases the risk of bacterial pneumonia [15,16,17,18,19]. Patients with OSA are found to have up to a 3-fold increased risk of pneumonia and a 5-fold increased risk of invasive pneumococcal disease [15,16,18]. The risk of pneumonia increases with OSA severity in a dose–response manner, with the highest risk observed in patients with severe OSA [17,18,19].

Hospitalized patients with pneumonia who have OSA tend to be about ten years younger and have a higher burden of comorbidities, such as obesity, chronic respiratory diseases, and heart failure, compared to patients without OSA [20]. Comorbidities, especially asthma and chronic obstructive pulmonary disease (COPD), as well as older age, increase the risk of LRTIs and LRTI recurrence in OSA patients [15]. While the relationship between OSA and pneumonia seems to be independent of baseline demographics and smoking, the confounding effects of obesity and multimorbidity are less clear. Most studies did not account for these [16] or have yielded mixed results [17,19]. In the Atherosclerosis Risk in Communities (ARIC) prospective cohort study with a median follow-up of 20.4 years, severe OSA increased the risk of pneumonia hospitalization by 87% after adjustment for baseline demographics and lifestyle, but this association was attenuated and became non-significant in models that included body mass index (BMI) or chronic respiratory conditions such as COPD and asthma [17]. Conversely, in a large retrospective cohort study from Taiwan, OSA was associated with a 20% increased risk of incident pneumonia independent of comorbidities including pre-existing diabetes mellitus, hypertension, coronary heart disease, heart failure, cerebrovascular disease, dementia, epilepsy, Parkinson’s disease, chronic kidney disease, liver cirrhosis, gastroesophageal reflux disease, cancer, asthma, COPD, and tuberculosis [19]. Notably, the ARIC study found that a hypoxic burden, represented by the time with oxygen saturation below 90% during total sleep time (T90) of more than 5% versus less than 1%, was associated with a 50% increased risk of pneumonia. This association was observed in models that accounted for BMI, suggesting that the hypoxic burden may capture the physiological consequences of OSA on LRTIs that extend beyond the effects of obesity alone [17].

Two out of six studies evaluated the association between OSA and severity of LRTIs [18,20] (Table 2). A significant moderate positive correlation between the apnea–hypopnea index (AHI) and the pneumonia severity index (PSI) levels was shown in a study by Chiner et al. [18]. A retrospective cohort study of patients with pneumonia at 347 United States hospitals showed that OSA is associated with the risk of more severe CAP, albeit with the same or lesser mortality. Patients with OSA were twice as likely to require invasive mechanical ventilation (IMV) at the time of admission and had more than a 50% increased likelihood of clinical deterioration requiring intensive care unit (ICU) admission and IMV later in the hospital stay [20]. The observed same or lower mortality compared to non-OSA patients may be related to the high prevalence of obesity in OSA and the so-called “obesity paradox,” by which obese patients may have a survival benefit likely due to increased metabolic reserve and more intense monitoring [43,44].

As OSA is associated with pneumonia severity, identifying OSA in pneumonia patients is important. However, the lack of somnolence, measured by the Epworth Sleepiness Scale in patients with OSA who have pneumonia [18], suggests that the level of daytime sleepiness is not a reliable way to screen for OSA in these patients. While the appropriateness of other questionnaires that, in addition to subjective symptoms, include anthropometric measurements such as STOP-Bang remains to be determined, performing a sleep study continues to be the preferred method to diagnose OSA in patients at risk for pneumonia.

## 4. Obstructive Sleep Apnea and Influenza Pneumonia

The influenza virus is highly transmissible. It can evade the immune system through antigenic drifts and shifts, leading to recurring seasonal epidemics and occasional pandemics [45,46]. Certain risk factors increase the severity of influenza infections. Respiratory diseases like asthma heighten the risk of hospitalization [47], and COPD is linked with increased morbidity and mortality [48]. Therefore, identifying patients at high risk for severe influenza is critical for promoting vaccination and adequate preventive measures.

The susceptibility to influenza infection among patients with OSA appears to be heightened, although the evidence is of poorer quality. Nonetheless, the current body of literature suggests that individuals with OSA are at a higher risk for severe manifestations of influenza. Two retrospective studies, investigating the comprehensive Taiwanese national database, indicated that OSA patients, identified by International Classification of Diseases (ICD) codes, have an 18% higher risk of contracting influenza and almost two times higher risk of experiencing severe acute respiratory infection defined as influenza infection necessitating hospitalization [24,25]. In both studies, OSA was associated with outcomes independent of obesity and other relevant comorbid diseases. However, it must be noted that these studies are limited by the lack of data on BMI and OSA severity.

Larger, multicentric studies focusing on risk factors for severe influenza infections and patient outcomes have identified OSA, along with central sleep apnea syndromes, as significant contributors. These conditions have been linked to a nearly tenfold increase in the likelihood of ICU admission [21] and a four times higher risk of non-invasive ventilation (NIV) failure [22] in patients with influenza infection. However, these studies did not find a statistically significant association between OSA and influenza mortality rates, suggesting that the presence of OSA exacerbates the course of influenza without necessarily affecting the outcome of the infection.

A retrospective cohort study by Mok et al. revealed that OSA patients non-adherent to continuous positive airway pressure (CPAP) therapy show an increased likelihood of hospitalization due to influenza, compared to those adherent to CPAP [23]. Further, the CPAP-compliant cohort displayed higher AHI values than their non-compliant counterparts, although not to a statistically significant extent. This suggests that CPAP adherence could protect against severe influenza despite more severe OSA. Other patient outcomes, such as length of stay and disease complications, did not significantly differ between groups. However, due to a limited sample size, this study might be underpowered to detect all relevant differences.

## 5. Obstructive Sleep Apnea and COVID-19 Pneumonia

Several risk factors exacerbate COVID-19-related disease severity and outcomes. Among the well-documented risk factors are obesity, diabetes, cardiovascular diseases, and metabolic syndrome [49,50]. OSA shares many common risk factors with COVID-19, and many studies have explored a possible association between the two. Our search identified five systematic reviews and meta-analyses that aimed to clarify this link.

The initial systematic review, published in February 2021, included only six papers with original data, five of which were case series and reports and suggested an increased risk of severe outcomes from COVID-19 in OSA patients [51]. A subsequent meta-analysis that evaluated 428 studies on poor prognostic factors for COVID-19 patients, highlighted OSA as a potential contributor. This analysis included seven studies presenting data on OSA with over 3800 subjects, suggesting that OSA more than doubles the risk for hospitalization [52]. Additionally, a meta-analysis evaluating European studies pointed out that OSA increases the risk for COVID-19-related hospitalization almost three times, but is not a predictor of intrahospital mortality [53]. These conclusions seem to be based on two studies included in the analysis, one examining OSA and the risk for COVID-19-related hospitalization, and one investigating COVID-19 hospital mortality. Two meta-analyses from 2021 and 2022 presented more conclusive evidence regarding the link between OSA and COVID-19 [54,55]. A meta-analysis from 2021 including 21 studies and over 50,000 COVID-19 patients concluded that OSA increases the risk not only for severe COVID-19 disease, need for ICU admission and mechanical ventilation, but also mortality [54]. A meta-analysis published in 2022 by Hu et al. confirmed OSA as a predictor of fatal outcomes in COVID-19 patients, increasing the risk by 50% [55].

Given the established link between OSA and progressive COVID-19 disease and adverse outcomes, our current research seeks to expand on these findings by focusing further on multicentric observational studies or those using large national or regional databases to provide a broader understanding of how OSA impacts the risk of contracting the SARS-CoV-2 virus, severity of COVID-19 disease, and associated mortality risks. Individual studies are presented in Table 2.

The evidence concerning the impact of OSA on the severity of COVID-19 is relatively conclusive. Of the 17 studies reviewed, 15 examined various indicators of COVID-19 severity, including hospitalization, ICU admission, length of stay, acute respiratory failure, and the need for intubation. Notably, 13 of these studies reported OSA as an independent predictor of these severe disease indicators (Table 2). The study by Cade et al. [30] linked OSA to hospitalization and severe outcomes, and the study by Mashaqi et al. [26] associated it with ICU admissions and longer hospital stays. In both studies, however, these relationships weakened after adjustments for BMI and comorbidities. Only two studies found no correlation between OSA and COVID-19 severity even before making such adjustments [38,40]. Regardless of whether OSA is an independent risk factor, it is evident that this patient population is at an increased risk for COVID-19 disease progression requiring inpatient or ICU admission, longer hospital stays, and complications.

A study by Pena Orbea et al. [31] evaluated sleep-related hypoxia measures and identified higher values of T90 and lower oxygen saturation, but not AHI, as independent predictors of severe COVID-19 and mortality. These findings suggest that hypoxemia, rather than sleep interruptions found in OSA patients, more significantly affects COVID-19 progression.

Data on COVID-19-related mortality remain inconsistent: six studies concluded that OSA does not increase mortality risk and another five reported OSA as an independent predictor of mortality (Table 2). Challenges in determining contributors to mortality include the widespread underdiagnosis of OSA, reliance on ICD codes for detecting OSA, “obesity paradox”, and already very high mortality in critically ill patients. 

In addition to classical indicators of disease severity, such as hospitalization and mortality rates, there is evidence that patients with OSA are also at increased risk of post-acute sequelae of SARS-CoV-2, also known as long COVID syndrome [56,57]. Patients with OSA may face significant challenges if they develop long COVID. Persistent fatigue, lack of energy, cognitive disturbances, and mental disorders are common in both conditions [58,59,60,61,62,63,64]. While it is unclear to what extent OSA contributes to the symptoms of long COVID, data suggest that treating previously undiagnosed OSA in patients with long COVID may lead to the resolution of symptoms in some patients [65]. Cardiovascular complications are also a concern, as long COVID’s association with symptoms like palpitations and chest pain [59] could heighten the already increased risk of cardiac complications among OSA patients [66].

## 6. Obstructive Sleep Apnea and Lower Respiratory Tract Infections: Pathophysiology

The etiopathogenic link between OSA and an increased susceptibility to LRTIs is likely complex and influenced by several factors. For clarity, the mechanisms that may lead to acute LRTIs in OSA patients can be grouped into three main categories: altered immunity, risk of aspiration, and the role of comorbidities common in this population as depicted in Figure 1. These mechanisms likely act synergistically rather than in isolation.

### 6.1. Altered Immunity

OSA is considered a pro-inflammatory condition [67], primarily due to intermittent hypoxia [68] and sleep fragmentation [69]. IH induces oxidative stress, leading to increased tumor necrosis factor-alpha (TNF-α) transcription [67,70] and higher interleukin 6 (IL-6) levels, which correlate with OSA severity [71]. Furthermore, IL-6 elevates C-reactive protein (CRP) levels, which, alongside TNF-α, is linked to sleep fragmentation and deprivation [67,72]. The resulting chronic low-grade inflammation leads to continuous immune system activation and gradual deterioration, known as immune senescence, increasing vulnerability to infections [73,74].

IH and resultant inflammation in OSA could affect pulmonary tissue directly. Overexpressed hypoxia-inducible factor (HIF)-1α, observed in COPD patients [75], might also be present in OSA [76], increasing infection risks through upregulated platelet-activating factor receptors that some bacteria use as receptors [75]. Additionally, sleep disruption independently impacts immune function by altering NK cells, T cell redistribution [77], immune cell proportions, and cytokine levels [78]. This could impact the response to vaccination [77], boosting humoral but impairing cellular immunity, which leads to higher infection risks despite adequate antibody titers [79,80,81,82]. Moreover, there is a well-established link between disrupted sleep patterns and an increased risk of pneumonia [83]. Next, hypercapnia in OSA patients [84], particularly those with comorbid COPD [85] or obesity hypoventilation syndrome (OHS) [86], reduces neutrophil phagocytic capabilities and pro-phagocytic cytokines [87]. Finally, IH could worsen outcomes by increasing the hypoxemic burden in patients with LRTI-related hypoxic respiratory failure.

### 6.2. Risk of Aspiration

Aspiration of oropharyngeal contents in OSA patients represents a significant potential mechanism contributing to LRTIs, particularly from bacterial agents. Several factors link OSA to an increased risk of aspiration, including an association with gastroesophageal reflux disease (GERD), impaired swallowing, oropharyngeal sensory deficits, and generation of negative intrathoracic pressure during sleep apnea episodes.

The relationship between OSA and GERD is well-established and appears to be bidirectional [88,89,90,91]. Pneumonia risk in these patients might be elevated due to GERD-related microaspiration [92]. Further, proton pump inhibitors create a less acidic gastric environment, delay gastric emptying, and may enhance bacterial colonization, increasing infection risk with aspiration [93].

OSA patients often have dysfunctional swallowing mechanisms, oropharyngeal sensory impairment, and alterations in deglutition muscle fibers [94,95,96]. Persistent, low-grade upper airway trauma causes sensory nerve damage, impairing the adductor reflex that prevents aspiration [94]. This sensory-motor impairment leads to swallowing–respiration discoordination, evidenced by increased latency in triggering the swallowing reflex [97].

Aspiration risk in OSA patients is elevated during apnea episodes due to negative intrathoracic pressure [3,98], which facilitates oropharyngeal content suction into the lungs. Studies have shown that OSA patients aspirate larger volumes at night compared to healthy individuals [99,100], potentially increasing bacterial load to the lower respiratory tract. CPAP therapy has been shown to reduce deglutition disturbances and prevent aspiration episodes in OSA patients [101].

### 6.3. The Role of Obesity and Other Comorbidities

Although OSA might be identified as a potential independent risk factor for severe LRTIs and adverse outcomes, it is crucial to consider the broader spectrum of risks faced by the OSA patient population.

Obesity is a significant risk factor for OSA, with BMI as a key predictor [102,103]. In critically ill patients, obesity increases the risks of complications like difficult airway management, acute respiratory distress syndrome, acute kidney injury, thromboembolic events, and infections [104]. Conversely, the “obesity paradox” suggests a protective effect against mortality in obese patients with LRTIs, possibly due to earlier presentation, higher hospital admission likelihood, better nutritional reserves, reduced lung injury, and altered immune response with less inflammation [105,106].

OSA is linked to numerous diseases, either causally or through common risk factors. Conditions such as diabetes mellitus [107], hypertension [108], coronary artery disease [109], heart failure [110], asthma [111], COPD [85], and end-stage kidney disease [112] are prevalent among OSA patients. These conditions increase the risk of severe disease and complications from LRTIs [113,114]. Cardiac events are significant complications for LRTI patients, and comorbid chronic heart disease may predispose OSA patients to these events [115]. Similarly, comorbid COPD is important as these patients often receive corticosteroids, increasing pneumonia risk [116]. Many studies confirmed OSA as an independent risk factor, often adjusted for chronic diseases as covariates. However, OSA patients who also suffer from these conditions might be at an even greater risk.

## 7. Obstructive Sleep Apnea and Lower Respiratory Tract Infections: Treatment

### 7.1. Settings of Care and Empiric Antibiotics

In community-acquired acute LRTIs, determining the causal agent clinically or radiologically is challenging. Thus, the American Thoracic Society and Infectious Diseases Society of America (ATS/IDSA) 2019 guidelines recommend empirical antibiotic therapy [117]. The choice of regimen depends on the care setting, complication risk, and illness severity, with clinical decision support tools like the pneumonia severity index (PSI) aiding hospitalization decisions [117,118]. OSA can affect PSI scores due to comorbid conditions like heart failure and kidney disease. Further, diabetes elevates glucose levels, while COPD and/or OHS contribute to hypercapnia, worsening acidosis, hypoventilation, and hypoxemia. All these factors may increase PSI scores, resulting in hospital admission and necessitating the use of broader antimicrobial coverage.

For outpatient management, ATS/IDSA recommends regimens based on complication risk [117]. Patients with chronic conditions like heart, lung, liver, or renal disease, or diabetes should receive dual therapy with an anti-pneumococcal beta-lactam (e.g., amoxicillin-clavulanate or cefuroxime) and a macrolide (azithromycin or clarithromycin) or doxycycline. Alternatively, respiratory fluoroquinolones (gemifloxacin, levofloxacin, or moxifloxacin) can be used as monotherapy. Given their higher risk of streptococcal pneumonia and invasive pneumococcal disease [16], OSA patients may benefit from this more aggressive approach compared to beta-lactam-, macrolide-, or doxycycline-monotherapy.

### 7.2. Specific Risks Guiding Empiric Antibiotic Therapy

Guideline-recommended antibiotic regimens for CAP cover common typical (*Streptococcus pneumoniae*, *Haemophilus influenzae*, *Staphylococcus aureus*, *Moraxella catarrhalis*) and atypical pathogens (*Mycoplasma pneumoniae*, *Legionella species*, *Chlamydia pneumoniae*) [117]. A study identified *Pseudomonas aeruginosa* in over 10% of OSA patients with bacterial pneumonia [18]. Conditions like chronic aspiration, COPD, coronary artery disease, heart failure, and PPI use [119], common in OSA patients, increase Pseudomonas risk. Frequent hospitalizations, antibiotic use, and ICU admissions also heighten this risk [119]. Thus, in severe LRTI cases, early coverage for Pseudomonas with an antipseudomonal beta-lactam or respiratory fluoroquinolone is advisable [117].

Case reports indicate OSA patients may develop Legionnaires’ disease from CPAP machines [120,121]. If *Legionella* is suspected, characterized by relative bradycardia, diarrhea, neurological disturbances, and/or hyponatremia [122], early testing for *Legionella* urine antigen and treatment with a respiratory fluoroquinolone over macrolides should be considered [123].

Historically, anaerobic coverage was recommended for patients prone to aspiration, like those with OSA [124]. However, current guidelines advise against routine anaerobic coverage, such as clindamycin or metronidazole, unless there is a lung abscess or empyema, as aspiration pneumonia bacteria are similar to CAP pathogens [117].

For influenza or COVID-19, the CDC recommends specific antiviral therapies for patients with chronic diseases such as asthma, COPD, and pulmonary fibrosis [125,126]. Given the increased risk and severity of infections in OSA patients, antivirals should also be considered in this population.

### 7.3. Antibiotic Pharmacokinetics, Side Effects, and Resistance

The pharmacokinetics of medications, including antibiotics, can be altered in OSA patients. IH inhibits cytochrome P450 expression in animal models [127,128], leading to impaired metabolism and higher drug concentrations, potentially increasing toxicity risk. This affects antimicrobials undergoing hepatic metabolism, such as macrolides [129]. Additionally, OSA-associated non-alcoholic fatty liver disease [130] can further slow drug metabolism. Further, antibiotics like vancomycin [131] may require dose adjustments in OSA patients with higher BMI. Finally, renal impairment necessitates careful dosage adjustments for many antibiotics due to altered renal clearance [132].

Attention should also be given to the side effects of pneumonia antibiotics in OSA patients with comorbidities. The cardiotoxic effects of fluoroquinolones [133] and macrolides [134], and the renal toxicity of vancomycin [135], pose significant risks to OSA patients with cardiac or renal diseases. OSA patients’ increased exposure to antibiotics due to a higher pneumonia risk can lead to antibiotic resistance [136], posing additional risks.

## 8. Discussion

The existing evidence indicates that in adults, OSA increases the risk and severity of both bacterial and viral pneumonia. However, the data on mortality are less consistent, showing either the same or decreased risk compared to non-OSA patients for CAP, while presenting an increased mortality risk in some, but not all, studies involving COVID-19 patients. OSA seems to have no significant effects on mortality in influenza patients, but this has not been extensively studied. It is unclear to what extent obesity and a high comorbidity burden mediate the observed association. Indeed, in some studies, the risk of pneumonia hospitalization in OSA patients was attenuated after adjustment for BMI and comorbidities.

Most of the available evidence comes from studies that used ICD codes to evaluate both OSA and the presence of comorbidities. While administrative data are commonly used for large datasets as it is easy to obtain and non-costly, they may introduce potential biases due to coding variability and missing data [137]. Consequently, the results of these studies should be interpreted with caution. The use of administrative data also lacks granularity; thus, OSA severity was commonly not reported, and OSA features such as sleep duration, sleep fragmentation, and hypoxic burden were not evaluated. This is important as both sleep fragmentation and hypoxemic burden are associated with oxidative stress and inflammation, but their relative contribution to pneumonia risk remains unknown. Only a few studies evaluated T90, showing that it was a better predictor of pneumonia severity when compared to AHI [17,31]. Other sleep disorders that result in sleep disruption such as insomnia or shift work-related circadian rhythm sleep disorder may also increase susceptibility to respiratory infections [83]. In addition, the clinical presentation of OSA is heterogeneous with a variety of clinical phenotypes that differ in symptoms and comorbidities, which cannot be captured solely by AHI severity.

While CPAP remains the first-line therapy for OSA, there have been some concerns about whether CPAP increases the risk of respiratory infections further. Infrequent cleaning, improper handling, and the use of humidification with CPAP can create an environment that allows for the growth of microorganisms and subsequently increases the risk of infections [138,139]. Some studies, as well as several case reports, have suggested that CPAP therapy may increase the risk of respiratory infections in CPAP users [120,121,140,141,142]. Therefore, regular cleaning and maintenance of CPAP equipment are necessary to mitigate these potential risks [143]. Despite these concerns, data from recent studies are reassuring. A retrospective case-control study by Mercieca et al. found no difference in the prevalence of upper and lower respiratory tract infections or nasal swab results between CPAP and non-CPAP-treated patients, irrespective of humidifier use [144]. Additionally, in a retrospective cohort study of 482 adult patients with OSA, CPAP use was not associated longitudinally with the risk of respiratory infections [145].

At the start of the COVID-19 pandemic, the medical community was unfamiliar with the factors increasing the risk of contracting the novel SARS-CoV-2 virus. Given the shared comorbidities between OSA and COVID-19 [146], OSA patients were considered at higher risk. These patients were inadvertently placed at even greater risk due to measures taken to “flatten the curve” to prevent the spread of this airborne infection [147,148]. Specifically, there were recommendations to cease the use of CPAP machines because of concerns about increased virus transmission through aerosols and droplets [149,150]. Despite these concerns, subsequent studies indicated that the benefits of adhering to CPAP therapy not only outweigh the potential risks but also protect against severe COVID-19 disease and adverse outcomes in compliant patients [30,151]. Moreover, findings from influenza studies further confirm the protective effects of CPAP in OSA patients [23]. Furthermore, the use of novel technologies can be used to detect early changes in the respiratory health of CPAP users. CPAP telemonitoring data from COVID-19 patients showed reduced adherence or complete cessation of CPAP use in days preceding a COVID-19 diagnosis [152]. This offers new possibilities for early detection of respiratory infections in patients with OSA.

Current CDC guidelines advocate for pneumococcal vaccination for individuals over 65 and those under 65 with conditions such as COPD, emphysema, and asthma [153], along with annual influenza vaccinations for everyone aged six months and older [154], emphasizing the importance of those with respiratory conditions like asthma, COPD, and cystic fibrosis [155]. Heightened vigilance against COVID-19 is also recommended, particularly in light of new variants, which exhibit significant evasion of immunity provided by vaccines [156,157,158]. Given the findings discussed in this review study, indicating that patients with OSA are at an increased risk of severe outcomes from these infections, it would be advisable to include OSA patients in those prioritized for vaccination. Serologic studies for OSA patients concerning influenza and COVID-19 vaccinations [79,80,81], offer reassuring data, suggesting a positive response to vaccination. However, there might be a possibility of an immune system imbalance favoring humoral over cellular responses, potentially compromising the effectiveness of the protection from vaccination.

The association between LRTIs and OSA should also be considered in the context of global health. Low-income and middle-income countries (LMICs) have the highest LRTI morbidity and mortality burden [159] but are also facing increasing rates of obesity [160]. Still, the data on the prevalence and natural history of OSA in LMICs are scarce [161]. Although epidemiological data are lacking, the estimated prevalence of OSA in LMICs seems comparable or higher than in developed countries [1]. Given the evidence on the association between OSA and LRTIs, improving access to diagnosis and treatment for patients with suspected sleep apnea in LMICs could offer additional benefits in preventing LRTIs and improving outcomes in resource-limited settings.

This paper has several limitations inherent to narrative review methodology such as potential incomplete coverage, lack of quantitative analysis, and positive results bias. Heterogeneity in study designs and the lack of explicit criteria for study selection may have also introduced some bias. Nevertheless, to our knowledge, no previous study has comprehensively evaluated evidence on the association between OSA and various acute respiratory infections. 

## 9. Conclusions

OSA increases the risk and severity of both bacterial and viral acute LRTIs, with no clear evidence of increasing pneumonia mortality. However, the relative contribution of chronic comorbidities to the risk and prognosis of LRTIs in OSA patients is not fully understood. Future research should focus on identifying OSA indices beyond AHI that could help predict pneumonia risk as well as potential risks and benefits of CPAP therapy. Strategies to minimize the risk of infection such as vaccination should be considered, especially for patients with severe OSA. Finally, OSA should be considered an important risk factor for LRTIs in national and international guidelines.

## Figures and Tables

**Figure 1 antibiotics-13-00532-f001:**
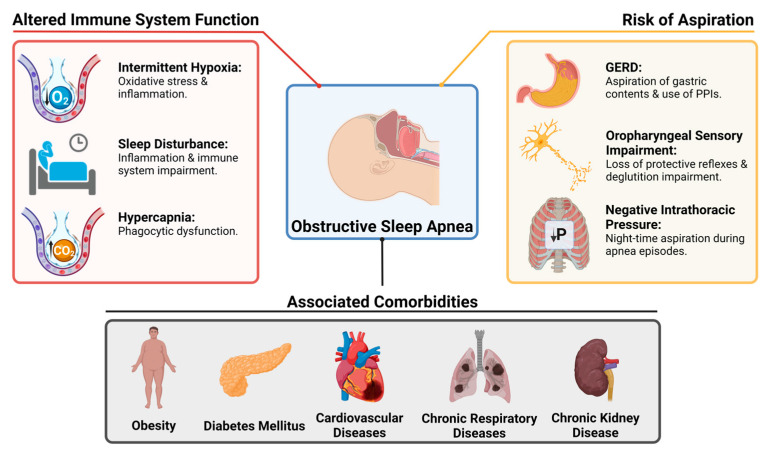
Pathophysiological Interactions and Risk Factors for Lower Respiratory Tract Infections in Patients with Obstructive Sleep Apnea. GERD: Gastroesophageal reflux disease, PPIs: Proton pump inhibitors.

**Table 1 antibiotics-13-00532-t001:** Keywords used for the search strategy of PubMed database.

(“Obstructive Sleep Apnea” OR “Sleep Apnea Syndromes” OR “Sleep-related breathing disorder” OR OSA) AND (pneumonia OR “acute pneumonia” OR “bacterial pneumonia” OR “community acquired pneumonia” OR CAP OR “lung infection” OR “respiratory infection” OR “bronchopneumonia”)
(“Obstructive Sleep Apnea” OR “Sleep Apnea Syndromes” OR “Sleep-related breathing disorder” OR OSA) AND (influenza OR “Influenza A” OR “Influenza B” OR “H1N1” OR “swine flu” OR “avian influenza” OR “H5N1” OR “seasonal influenza” OR “viral pneumonia” OR flu)
(“Obstructive Sleep Apnea” OR “Sleep Apnea Syndromes” OR “Sleep-related breathing disorder” OR OSA) AND (COVID-19 OR “SARS-CoV-2” OR “2019-nCoV” OR “coronavirus disease 2019” OR “novel coronavirus” OR “viral pneumonia”)

**Table 2 antibiotics-13-00532-t002:** The summary of the most relevant literature on the association between OSA and LRTIs.

Author and Date	Design	Total N (OSA N)	Inclusion and Exclusion Criteria	Outcomes	Key Findings	Limitations
Keto et al., 2023 [15]	Case-control from Finland	50,648 (25,324)	I: ICD code for OSA. E: OSA in the two years preceding the index date.	LRTI, recurring LRTI.	↑ LRTI in the year preceding OSA RR 1.35, and during the year after OSA RR 1.39.	No PSG data, no data on OSA treatment, no BMI data.
Grant et al., 2023 [16]	Retrospective cohort from healthcare plans database	38.62M PY (1.29M PY)	I: Minimum 1 year of enrollment in health plan. E: Death date before January 1st of the index year; Overlapping pneumonia inpatient admissions.	All-cause pneumonia, invasive pneumococcal disease, pneumococcal pneumonia.	OSA: ↑ pneumonia (18–49 y RR 3.6, 50–64 y RR 3.6, ≥65 y RR 3.4), ↑ invasive pneumococcal disease (18–49 y RR 5.7, 50–64 y RR 4.2, ≥65 y RR 4.2).	No PSG data, no data on OSA treatment, no BMI data.
Lutsey et al., 2023 [17]	Post-hoc analysis of the multicentric prospective cohort	1586 (772)	I: Valid PSG data; Self-identify as White. E: CSA; Already had the outcome of interest at the time of visit.	Hospitalization: with pneumonia; with respiratory infection; with any infection.	OSA not linked to outcomes; T90 > 5% ↑ hospitalized pneumonia HR 1.59, ↑ hospitalized respiratory infection HR 1.53, ↑ hospitalized any infection HR 1.25.	No data on OSA treatment, mostly White population.
Chiner et al., 2016 [18]	Single center case-control	123 (85)	I: Cases: Hospitalized for CAP; Controls: Hospitalized for non-respiratory/non-ENT infection. E: Previous OSA diagnosis and CPAP.	Pneumonia, PSI.	AHI ≥ 10: ↑ pneumonia OR 2.86; AHI ≥ 30: ↑ pneumonia OR 3.184; AHI positively correlated with PSI.	Small sample size, no data on OSA treatment.
Su et al., 2014 [19]	Retrospective cohort from Taiwan	34,100 (6816)	I: ICD codes for OSA; E: ICD codes for pneumonia, lung abscess, empyema.	Pneumonia.	OSA: ↑ pneumonia HR 1.19; OSA requiring CPAP: ↑ pneumonia HR 1.32.	No PSG data, no BMI data.
Lindenauer et al., 2014 [20]	Multicenter, retrospective cohort	250,907 (15,569)	I: ICD code for pneumonia; Chest radiography; Antibiotics within 48 h of admission. E: Transfers; Hospital LOS under 2 days; Cystic fibrosis; Pneumonia not present at admission.	ICU, MV, hospital mortality, hospital LOS, costs.	OSA: ↑ ICU OR 1.54, ↑ MV OR 1.68, ↑ hospital LOS RR 1.14, ↑ cost RR 1.22, ↓ mortality OR 0.90.	No PSG data, no data on OSA treatment, no BMI data.
Beumer et al., 2019 [21]	Two center, retrospective cohort	199 (9)	I: Symptoms and positive influenza PCR; Transfers if not received antibiotics or antivirals.	ICU, ICU mortality.	OSA/CSA: ↑ ICU admission OR 9.73., not linked to mortality.	Small sample size, no PSG data, no data on OSA treatment.
Boattini et al., 2023 [22]	Post-hoc analysis of a multicentric, retrospective cohort	356 (23)	I: Positive influenza or RSV PCR; Symptoms; Pulmonary infiltrate on imaging. E: Viral co-infections.	NIV failure, hospital mortality.	OSA/OHS: ↑ NIV failure OR 4.66, not linked to mortality.	No PSG data, no data on OSA treatment, no BMI data, no adjustments for obesity.
Mok et al., 2020 [23]	Single center, retrospective cohort	53 (53)	I: ICD codes for OSA, influenza. E: No PSG data; No OSA treatment data; CSA on PSG.	Hospitalization, complications, hospital LOS.	OSA non-CPAP vs. CPAP: ↑ hospitalization OR 4.7. Severity of OSA not linked to hospitalization in CPAP-non adherent.	Small sample size, no adjustments for obesity and comorbidities.
Tsai et al., 2022 [24]	Retrospective cohort from Taiwan	32,540 (6508)	I: Cases: ICD codes for OSA; Controls: No OSA; Randomly selected, matched by income, gender, urbanization, and age. E: influenza pneumonia before OSA.	Influenza-associated SARI.	OSA: ↑ influenza-SARI HR 1.98, ↑ cumulative incidence of influenza-SARI.	No PSG data, no data on OSA treatment, no BMI data.
Chen et al., 2021 [25]	Retrospective cohort from Taiwan	27,501 (5483)	I: Cases: ICD codes for OSA; Controls: No OSA; Randomly selected, matched by age, sex, index years, and comorbidities. E: UPPP; influenza before OSA.	Influenza, composite (pneumonia, hospitalization).	OSA: ↑ influenza HR 1.18, ↑ pneumonia or hospitalization 1.79.	No PSG data, no data on OSA treatment, no BMI data.
Mashaqi et al., 2021 [26]	Multicentric, retrospective cohort	1738 (139)	I: Hospitalized; ICD codes, PSG report, self-report, STOP-BANG for OSA; ICD codes COVID-19. E: ICD for CSA and unspecified sleep apnea.	MV, ICU, hospital mortality, hospital LOS.	OSA not linked to ICU admission, hospital LOS, MV, or mortality.	No PSG data, no data on OSA treatment.
Maas et al., 2021 [27]	Multicentric, retrospective cohort	5544,884 (~44,877)	I: All patient encounters; January to June 2020.	COVID-19, hospitalization, respiratory failure.	OSA: ↑ COVID-19, OR 8.6, ↑ hospitalization, OR 1.65, ↑ respiratory failure, OR 1.98.	No PSG data, no data on OSA treatment.
Strausz et al., 2021 [28]	Retrospective cohort from FinnGen biobank	445 (38)	I: All positive COVID-19 PCR from FinnGen biobank.	Hospitalization, COVID-19.	OSA not linked with COVID-19, ↑ hospitalization, OR 2.93. Link attenuated after adjustment for BMI in meta-analysis.	Small sample size, no PSG data, no data on OSA treatment.
Rögnvaldsson et al., 2022 [29]	Retrospective cohort from Iceland	4756 (185)	I: Positive COVID-19 PCR. E: Nursing home; COVID-19 during hospitalization or rehabilitation.	Composite (hospitalization, mortality).	OSA: ↑ composite outcome (hospitalization and mortality) OR 2.0. OSA and CPAP: ↑ composite outcome (hospitalization and mortality) OR 2.4.	No PSG data for the control group, no BMI data for 30% of controls and 2% of the OSA group.
Cade et al., 2020 [30]	Multicentric, retrospective cohort	4668 (443)	I: Positive COVID-19 PCR; A minimum of two clinical notes, two encounters, and three ICD diagnoses.	Mortality, composite (mortality, MV, ICU), hospitalization.	OSA or CPAP not linked with mortality, MV, ICU, and hospitalization.	No PSG data, no data on OSA treatment.
PenaOrbea et al., 2021 [31]	Multicentric, retrospective control and case-control	5402 (2664)	I: Positive COVID-19 PCR; PSG record available.	COVID-19, WHO-designated COVID-19 clinical outcomes, composite (hospitalization, mortality).	AHI, T90, SaO_2_, ETCO_2_ and CPAP not linked with COVID-19. T90 and SaO_2_: ↑ WHO-designated COVID-19 outcomes ↑ hospitalization, ↑ mortality.	Included only patients who had indications for PSG.
Oh et al., 2021 [32]	Retrospective cohort from South Korea	124,330 (550)	I: ICD codes for COVID-19, chronic respiratory diseases. E: COVID-19 still hospitalized as of June 26, 2020.	COVID-19; hospital mortality.	OSA: ↑ COVID-19, OR 1.65, not linked to mortality.	No PSG data, no data on OSA treatment, no BMI data.
Gottlieb et al., 2020 [33]	Retrospective cohort from Chicago, IL.	8673 (288)	I: Positive COVID-19 PCR. E: Interhospital transfers.	Hospitalization, ICU.	OSA not linked to hospitalization, ↑ ICU, OR 1.58.	No PSG data, no data on OSA treatment.
Kendzerska et al., 2023 [34]	Retrospective cohort from Ontario, CA.	4,912,229 (324,029)	I: Alive at the start of the pandemic; Followed until March 31, 2021, or death.	COVID-19, ED, hospitalization, ICU, 30-day mortality.	OSA: ↑ COVID-19, csHR 1.17, ↑ ED, csHR 1.62, ↑ hospitalizations csHR 1.50, ↑ ICU csHR 1.53, not linked to mortality.	No PSG data, no data on OSA treatment, no BMI data.
Peker et al., 2021 [35]	Multicenter, prospective, observational clinical trial	320 (121)	I: Positive COVID-19 PCR and/or clinical/radiologic.	Clinical improvement, clinical worsening, hospitalization, oxygen, ICU.	OSA: ↑ delayed clinical improvement, OR 0.42, ↑ oxygen OR 1.95, ↑ clinical worsening.	No PSG data, no data on OSA treatment.
Girardin et al., 2021 [36]	Retrospective cohort from NYC and LI	4446 (290)	I: Positive COVID-19 PCR.	Hospital mortality.	OSA not linked to mortality.	No PSG data, no data on OSA treatment, no BMI data.
Gimeno-Miguel et al., 2021 [37]	Retrospective cohort from Aragon, ES.	68,913 (1231)	I: Positive COVID-19 PCR/antigen; E: Patients diagnosed from March to May 2020.	Composite (hospitalization, 30-day mortality)	OSA: ↑ composite outcome (hospitalization and 30-day mortality) in women OR 1.43, but not in men.	No PSG data, no data on OSA treatment, no BMI data.
Cariou et al., 2020 [38]	Multicentric, retrospective cohort	1317 (114)	I: Positive COVID-19 PCR or clinical/radiological diagnosis, hospitalized, diabetics.	Composite (MV, 7-day mortality), mortality on day 7, MV on day 7, ICU, discharge on day 7.	OSA: ↑ mortality by day 7 OR 2.80, not linked to composite outcome (intubation and death within 7 days of admission).	No PSG data, no data on OSA treatment, diabetic population.
Ioannou et al., 2020 [39]	Longitudinal cohort from VA registry.	10,131 (2720)	I: VA enrollees who had COVID-19 PCR test; E: VA employees.	Hospitalization, MV, mortality.	OSA: ↑ MV HR, 1.22, not linked to hospitalization, mortality.	No PSG data, no data on OSA treatment, male veterans.
Izquierdo et al., 2020 [40]	Multicentric, retrospective cohort	10,504 (212)	I: Positive COVID-19 PCR or clinical/radiological diagnosis.	ICU.	OSA not linked to ICU admission.	No PSG data, no data on OSA treatment, no BMI data, no adjustments for obesity and comorbidities.
Lohia et al., 2021 [41]	Multicentric, retrospective cohort	1871 (63)	I: Adults; Positive COVID-19 PCR; E: Readmission; Ambulatory surgery, pregnant, transferred-for-ECMO patients.	Mortality, MV, ICU.	OSA ↑ mortality OR 2.59, ↑ ICU OR 1.95, ↑ MV OR 2.20.	Small OSA sample size, no data on OSA treatment, mostly African Americans.
Prasad et al., 2024 [42]	Retrospective cohort from VA registry	20,357 (6112)	I: Tested for COVID-19 by PCR; Until 16 December 2023.	COVID-19, LFNC, HFNC, NIV, MV, 30-day readmission; hospital LOS, ICU LOS, adapted WHO severity scale.	OSA ↑ COVID-19 OR 1.37, ↑ NIV OR 1.83, not linked to LFNC, HFNC, MV, 30-day readmission. CPAP adherence not linked to outcomes.	No PSG data.

Footnotes: AHI: apnea–hypopnea index, BMI: body mass index, CAP: community-acquired pneumonia, CA: Canada, CPAP: continuous positive airway pressure, CSA: central sleep apnea, csHR: cause-specific hazard ratio, E: exclusion criteria, ECMO: Extracorporeal Membrane Oxygenation, ED: emergency department, ENT: ear, nose, throat, ES: Spain, ETCO_2_: end-tidal CO_2_, HFNC: high-flow nasal cannula, HR: hazard ratio, I: inclusion criteria, ICU: intensive care unit, ICD: international classification of diseases, IL: Illinois, LI: Long Island, LOS: length of stay, LRTI: lower respiratory tract infection, LFNC: low-flow nasal cannula, M: million, MV: mechanical ventilation, N: number of subjects included, NYC: New York City, NIV: non-invasive ventilation, OHS: obesity hyperventilation syndrome, OR: odds ratio, OSA: obstructive sleep apnea, PCR: polymerase chain reaction, PSG: polysomnography, PSI: pneumonia severity index, PY: person-years, RSV: respiratory syncytial virus, RR: relative risk, SaO_2_: oxygen saturation, SARI: severe acute respiratory infection, STOP-BANG: screening tool for predicting obstructive sleep apnea, T90: percentage of sleep time with oxygen saturation below 90%, UPPP: uvulopalatopharyngoplasty, VA: Veterans Affairs, vs.: versus, WHO: World Health Organization, ↑: increased risk, ↓: decreased risk, (OSA N): number of study participants with obstructive sleep apnea, ~: approximation based on presented data. For studies covering COVID-19, only observational retrospective or prospective multicentric studies or ones leveraging large regional or national databases were included. The key findings column contains results mostly from multivariable models adjusting for various covariates.

## Data Availability

No new data were created.

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
