# Peer review of "Obstructive Sleep Apnea and Acute Lower Respiratory Tract Infections: A Narrative Literature Review"

_antibiotics, 2024, doi:10.3390/antibiotics13060532_

Round 1

Reviewer 1 Report

Comments and Suggestions for Authors

This is an excellent review and update on the risk of pneumonia in patients with sleep disordered breathing, specifically obstructive sleep apnoea.

The review is comprehensive, the methodology excellent and allows the reader to draw strong conclusions about the relationship between sleep apnoea and lower respiratory tract infections. The authors should suggest that sleep apnoea be included among the risk factors for pneumonia in national and international guidelines.

Reviewer 2 Report

Comments and Suggestions for Authors

Reviewer 3 Report

Comments and Suggestions for Authors

This is a narrative review that aims to evaluate the current evidence on the association of obstructive sleep apnea with the incidence and outcomes of acute lower respiratory tract infections in adults, specifically between OSA and community-acquired pneumonia (CAP), as well as influenza and COVID-19 pneumonia.

The authors have done a good job of gathering a large body of evidence from the literature. However, they have failed to present the results coherently and systematically.

The following issues require the attention of the authors:

1. Table 2 must include the number of patients with OSA, in addition to the total number of participants in each study.

2. It is imperative to clearly state the number of studies reporting specific outcomes, such as the risk of pneumonia in patients with OSA.

3.In the results section, especially in the subsection "Obstructive Sleep Apnea and Community-Acquired Pneumonia," all outcomes should be accompanied by odds ratios and the number of participants in the respective studies.

Round 2

Reviewer 3 Report

Comments and Suggestions for Authors

In the revised submission, the authors have successfully addressed all concerns raised in my previous review.